# Untargeted Metabolite Profiling for Screening Bioactive Compounds in Digestate of Manure under Anaerobic Digestion

**Jiaxin Lu** [1], **Atif Muhmood** [1], **Wojciech Czekała** [2], **Jakub Mazurkiewicz** [2], **Jacek Dach** [2] and **Renjie Dong** [1,*]

1   College of Engineering, China Agricultural University, Beijing 100083, China;
    bs20173070551@cau.edu.cn (J.L.); atif@cau.edu.cn (A.M.)
2   Institute of Biosystems Engineering, Poznań University of Life Sciences, Wojska Polskiego 50, 60-627 Poznań,
    Poland; wojciech.czekala@up.poznan.pl (W.C.); jakub.mazurkiewicz@up.poznan.p (J.M.);
    jacek.dach@up.poznan.pl (J.D.)
*   Correspondence: rjdong@cau.edu.cn; Tel.: +86-10-6273-7963

**Abstract:** Untargeted metabolite profiling was performed on chicken manure (CHM), swine manure (SM), cattle manure (CM), and their respective digestate by XCMS coupled with MetaboAnalyst programs. Through global chemical profiling, the chemical characteristics of different digestates and types of manure were displayed during the anaerobic digestion (AD) process. As the feed for AD, CM had less easily-degradable organics, SM contained the least O-alkyls and anomerics of carbohydrates, and CHM exhibited relatively lower bio-stability. The derived metabolite pathways of different manure during the AD process were identified by MetaboAnalyst. Twelve, 8, and 5 metabolic pathways were affected by the AD process in CHM, SM, and CM, respectively. Furthermore, bioactive compounds of digestate were detected, such as amino acids (L-arginine, L-ornithine, L-cysteine, and L-aspartate), hormones (L-adrenaline, 19-hydroxy androstenedione, and estrone), alkaloids (tryptamine and N-methyltyramine), and vitamin B5, in different types of manure and their digestates. The combination of XCMS and MetaboAnalyst programs can be an effective strategy for metabolite profiling of manure and its anaerobic digestate under different situations.

**Keywords:** anaerobic digestion; bioactive compounds; metabolites; untargeted profiling; metabolic pathways

## 1. Introduction

With the rising demand for limited waste disposal and renewable energy, anaerobic digestion technology has attracted extensive attention and shows advantages [1]. In recent years, the number of biogas plants has been increased rapidly, which has led to enormous environmental and economic benefits to the communities in rural areas [2]. Consequently, large amounts of anaerobic digestate have been generated as a by-product in anaerobic digestion (AD) plants during biogas production. Generally, the digestate is rich in macronutrients (e.g., N, P, K) and is traditionally used as fertilizers in agriculture. It has been well documented that the supplementary application of digestate to wheat, maize, grass, pasture, peanut, green pepper, eggplant, cabbage, cucumber, autumn lettuce, and sorghum can improve the plant growth, yield, and quality [3–7]. In addition to supplying nutrients to the plants, the application of digestate has been reported to play an essential role in the inhibition of plant pathogens (fungal and bacterial) and insects [8]. For several decades, farmers have observed that the land application of the digestate promotes disease resistance in plants [9]. However, most of the current publications only record superficial observations. The underlying mechanism of promoting disease resistance by the digestate is not well understood. Several studies have speculated that the

digestate promotes disease resistance through bioactive compounds, which exhibit antimicrobial and antifungal activities against soil pathogens. However, there is very little progress in identifying and characterizing these bioactive molecules in the digestate, which may be due to the complex properties of the manure digestate.

The degradation of organic compounds under anaerobic conditions results in the formation of recalcitrant fractions like biopolymers, steroids, and lignin [10]. These recalcitrant compounds have been reported to be the precursors for humus formation [11]. Similarly, several studies have reported that bioactive compounds, such as fatty acid derivatives, alkaloids, flavonoids, and terpenoids originate from the secondary metabolites generated during the digestion of organic matter, which has been partially attributed to the disease resistance property of the digestate [12]. However, the composition of the digestate is greatly affected by the feedstock and the operation condition of the biogas plants. Additionally, the identification and characterization of bioactive molecules during AD is limited by the need for advanced analytical techniques. The biological activity of the digestate is the result of multiple metabolite interactions, as far as the metabolomic profile is concerned. It is difficult to explain the functional performance of the digestate with conventional quantitative analyses.

Untargeted metabolic fingerprinting of gas chromatography-mass spectrometry (GC-MS) or liquid chromatography-mass spectrometry (LC-MS) data coupled with chemometrics has proven to be a robust tool for multiplex metabolomic profiling from a global perspective [13,14]. In a typical analysis, tens of thousands of metabolites can be measured by LC-MS in the anaerobic digestate of manure [15]. The untargeted metabolomic analysis allows for an assessment of the metabolites extracted from a sample and can reveal novel and unanticipated perturbations in an unbiased means to examine the relationship between interconnected metabolites from multiple pathways [16]. Differing reproducibility criteria, routinely used in the quantitative analyses, non-targeted screening, emphasized in the evaluation of the joint multi-platform analysis, increase the accuracy and ultimately reproducibility in order to achieve global profiling of the compounds in the environmental sample and rank them in terms of their environmental significance [17]. Previously, the untargeted metabolic analysis has been employed for a wide range of applications, including cancer research, chronic pain, long-term studies of human serum, fruit and bud development and stem cell differentiation in plants [18–20].

The aims of this study were to analyze the untargeted metabolomic profiling and meta-analysis data of biochemical changes that occur during AD of three different kinds of manure, based on the cloud-based mass spectrometric data analysis software (XCMS) integrated with the MetaboAnalyst program. The specific objectives of the study were: identification of metabolites generated during the AD of manure, investigation of the metabolic pathways responsible for metabolite generation, identification and quantification of bioactive compounds derived from secondary metabolism during the AD of various manures.

## 2. Materials and Methods

### 2.1. Materials

The anaerobic digestate of chicken manure (CHM), cow manure (CM), and swine manure (SM) was collected from the Liu Minying (39.7° N, 116.6° E), Doudian (39.7° N, 116.6° E) and Donghuashan (40.2° N, 116.9° E) large-scale biogas plants, located in the suburbs of Beijing, China. The biogas digesters were chosen based on the similarity in operating conditions, including temperature (mesophilic) and organic loading rate (OLR), with different feedstocks to assess the variability in metabolite generation during AD of different feedstocks. We also collected raw manure samples (CHM, CM, and SM) for analysis. The three large-scale biogas plants have operated continuously for more than 10 years with the single feedstock. After the fresh digestate was discharged from each plant, the samples were collected immediately after stirring. All manure and digestate samples were analyzed for physicochemical characteristics such as volatile solids, chemical oxygen demand, ammonical and nitrate nitrogen

following the standard methods of the American Public Health Association Standard (APHA, 2012). pH was determined using a digital pH meter (FE20, METTLER TOLEDO, Greifensee, Switzerland), while ammonical nitrogen was measured following the phenate method using a UV spectrophotometer (Shimadzu UV-1800, Kyoto, Kyoto Prefecture, Japan). The AD process parameters and the chemical composition of the raw manure samples and their anaerobic digestate are given in Table 1.

**Table 1.** The biogas plant parameters and characteristics of manure and respective digestate (n = 3).

| Biogas Plant | Doudian | | Donghuashan | | Liu Minying | |
|---|---|---|---|---|---|---|
| Model | CSTR | | USR | | USR | |
| Temperature (°C) | 33–35 | | 32–35 | | 30 | |
| HRT (d) | 30 | | 20 | | 15 | |
| OLR (kg m$^{-3}$d$^{-1}$) | 3–4 | | 4.0–4.7 | | 3–4 | |
| **Parameters** | **Cow Manure** | **Digestate** | **Swine Manure** | **Digestate** | **Chicken Manure** | **Digestate** |
| pH | 6.80 ± 0.18 | 7.75 ± 0.23 | 6.55 ± 0.20 | 7.41 ± 0.18 | 7.78 ± 0.23 | 8.38 ± 0.26 |
| VS (%) | 7.20 ± 0.27 | 6.03 ± 0.02 | 7.64 ± 0.22 | 6.33 ± 0.33 | 7.10 ± 0.59 | 4.02 ± 0.68 |
| COD (g L$^{-1}$) | 3.20 ± 1.03 | 8.8 ± 4.32 | 9.4 ± 2.23 | 18.6 ± 3.34 | 78.7 ± 3.56 | 67.5 ± 6.89 |
| NH$_4^+$-N (mg L$^{-1}$) | 333.4 ± 62 | 1959 ± 79 | 979.3 ± 29.3 | 1938 ± 7.80 | 8199 ± 53.3 | 7032 ± 34.6 |
| NO$_3^-$-N (mg L$^{-1}$) | 1.70 ± 0.27 | 0.89 ± 0.13 | 2.02 ± 0.29 | 0.96 ± 0.14 | 6.71 ± 0.45 | 4.71 ± 0.92 |

CSTR = continuous stirred tank reactor, USR = up-flow solid reactor, HRT = hydraulic retention time, OLR = organic loading rate, VS = volatile solids, COD = chemical oxygen demand, NH$_4^+$-N = ammonical nitrogen, NO$_3^-$-N= nitrate nitrogen.

### 2.2. Extraction and GCMS Detection

The anaerobic digestate of CHM, SM, and CM were extracted and concentrated before the samples were subjected to GC-MS. Approximately 15 mL of anaerobic digestate of CHM, CM, and SM was mixed separately with 15 mL dichloromethane (1:1) and added to 10g NaCl. The sample was incubated on an oscillator for 1 h and then extracted using an ultrasonic device (KH300SP, 25 kHz, 300 W, Kunshan Ultrasonic Instrument Co., kunshan, China) for 20 min. After centrifuging at 6000 rpm for 30 min, the supernatant of the samples underwent vacuum distillation using a lab-scale vacuum distillation system (Heidolph Laborota 4001, Sigma-Aldrich, Saint Louis, MO, USA) under a vacuum (rotating temperature, 40 °C; relative pressure, 96 kPa, equivalent to an absolute pressure of 5.3 kPa). The concentrated sample was stored in a closed environment at 4 °C until analysis. All experiments were repeated three times to obtain statistically reliable results.

The extracts were injected into GC-MS using a gas chromatography-mass spectrometer (Agilent 7890, Agilent Technologies, Inc. Santa Clara, CA, USA) equipped with a DB-5MS capillary column (30 m length, 0.25 mm inner diameter, 0.25 μm film thickness) and a quadrupole analyzer. The mass range we used for our analysis was 35 to 600 m/z. About 1.5 μL of the extract was injected manually into the instrument using a microsyringe for ion chromatographic analysis as described by Lu, Muhmood [21]. The relative content of the constituents of each extract was expressed as a percentage with peak area normalization. The compounds were identified by comparing the mass spectra with NIST08 library data.

### 2.3. Data Analysis

The manure and digestate samples were randomized across the whole data acquisition sequence to prevent biased statistical analyses or biased metabolomic data structure by non-biological factors such as instrumental drift. Comparative profiling was performed in MS mode. The data were processed using XCMS Online (xcmsonline.scripps.edu) to identify the metabolite of interest that exhibited a statistically significant difference. XCMS is a cloud-based informatics platform that can conveniently process and visualize mass-spectrometry-based untargeted metabolomic data and perform statistical analysis and metabolite assignment, which can be used for biological interpretation [22,23].

The raw GC-MS data (wiff.scan files) were exported from Chemstation as "AIA" and processed by XCMS. Uploaded data were used for metabolite feature detection and retention time correction and alignment. Peaks were matched across all the samples filtered through 0.01 m/z and 0.5 min tolerance. The normalized data were downloaded for multivariate analysis. The manure types (CHM, CM, and SM) and their corresponding anaerobic digestate were analyzed as a pairwise job to perform an unpaired parametric t-test (Welch test) with the following settings: wave feature detection at 2.5 ppm with maximum tolerance in consecutive scans, and obi wrap retention time correction with 1 m/z step size (profited). The statistically significant features were identified in the digestate of CHM, CM, and SM using p-value $\leq 0.05$ and threshold $\geq 1.5$. The generated files were imported as Meta XCMS for multivariate analysis. The meta XCMS analysis compares the metabolomics datasets containing any number of sample groups that are unique or shared among all or some of the treatments [23]. No restrictions were imposed to detect up-regulation or down-regulation of metabolite features. The second-order comparison was applied using a tolerance of 0.01 m/z and 60 s retention time. The significant metabolite features were identified through MS/MS matching to standards in the metabolite METLIN Database.

The metabolic features of three different independent paired comparisons between manure types and their anaerobic digestate containing m/z, *p*-value, and t score were imported to the MetaboAnalyst 4.0 software (Wishart Research Group, University of Alberta, Alberta, Canada) [24] to generate the related derivative pathway impact. The pathway analysis module integrates results from enrichment analysis and topology analysis based on the Kyoto Encyclopedia of Gene and Genome (KEGG) and human Metabolome Database (HMDB). The importance of a compound within a given metabolic network was estimated by its centrality measure containing "degree" centrality and "betweenness" centrality. Further information on graph-based methods can be obtained elsewhere [25].

## 3. Results and Discussion

### 3.1. Primary Metabolite Profiling

The change in the metabolite profile after AD of CHM, CM, and SM is presented in a cloud plot (Figure 1). XCMS-generated cloud plots allow an effective representation of GC-MS-based metabolomics data by providing information including the p-value, the directional fold change, the retention time, and the mass-to-charge ratio of metabolic features within a defined threshold [22,26]. The metabolite features were defined as ions with unique m/z and retention time values. The tentative identification of metabolite features was performed by matching the m/z measurements with the METLIN database. As shown in Figure 1, the up-regulated (green spots) metabolites exhibited a maximum fold change at the retention time of 20 to 40 min with a mass-to-charge (m/z) range of 0 to 200. Moreover, the down-regulated features were observed to be more intense at a retention time range of 15–20 min and 30–35 min with a maximum fold change for CM and CHM, respectively. A total of 479, 241, and 431 metabolites were found to be significantly ($p < 0.05$) altered between CHM and its digestate, CM and its digestate, and SM and its digestate, respectively. Furthermore, there were 272, 176, and 108 down-regulated ions and 207, 65, and 323 up-regulated ions between CHM and its digestate, CM and its digestate, and SM and its digestate, respectively. CM had the lowest number of significant metabolites during AD because the CM had less easily-degradable organics than SM and CHM [27]. The number of metabolites in CHM was higher than that in its digestate. The number of down-regulated metabolites was higher than the number of newly generated metabolites after AD of CHM, which was due to the relatively lower bio-stability of CHM and resulted in complex composition and change in the metabolite profile during AD compared to CM and SM [27]. The number of newly generated ions was significantly ($p < 0.05$) higher than the number of decomposed ions after AD of SM. We observed a significant difference ($p < 0.05$) in the metabolite profile among all manure types and their digestates. Most of these up-regulated features and down-regulated metabolites have corresponding database hits (spots with a black outline).

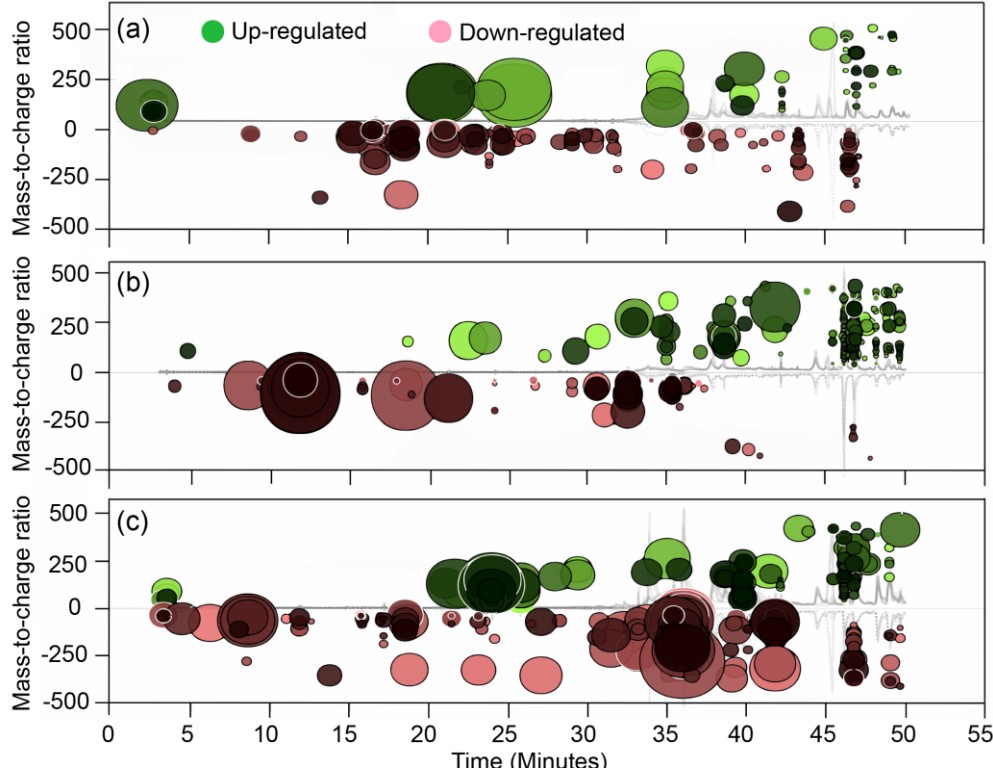

**Figure 1.** XCMS Cloud plot representation of the significant characteristic features (metabolites) between (**a**) cow manure and digestate, (**b**) swine manure and digestate, (**c**) chicken manure and digestate. Green colored spots correspond to features generated from up-regulated metabolites indicating the metabolite features whose intensity increased after AD. Red-colored spots represent the features generated from down-regulated metabolites indicating the metabolite features whose intensity decreased after AD. The size of each circle represents the fold change of feature changes. The p-values are represented by the color intensity of the metabolite feature, where the features with high p-values are darker compared to the features with low *p*-values. The 'y' coordinate of each feature corresponds to the mass-to-charge ratio of the compound.

A meta-analysis was conducted to identify the marker compounds that were responsible for the change in metabolite profile after AD (Figure 2). The analysis revealed that about 708, 296, and 344 characteristic features were altered after AD of CHM, CM, and SM, respectively. The number of metabolites altered during AD of manure from CHM (708) and SM (344) was high. Thirty-six shared metabolites were observed among all three manures, including 30 identified metabolites (Table S1). The shared metabolites that were significantly changed during the AD process of CHM, CM and SM included fatty acids, amino alcohols, amides, amino esters, amines, isothiocyanate, volatile gases such as 1-pentanethiol, hydrogen chloride, and feed additives such as sodium pantothenate.

Furthermore, HCA (heat chart analysis) was performed to evaluate the content of the 36 common metabolites among individual manures and the digestate based on the ranking of significant metabolites (Figure 3). The analysis revealed that 15 metabolites were up-regulated and 15 were down-regulated in CHM and CM after digestion, while only six metabolites (2-methyloxy-1,1,1 triamine hexamethyl ethane, 1-pentanamine hydrofluoride, lithium cyclohexyl, hexane, 1-aminopentane, and *N*,*N*-1,1-tetramethyl boranamine) were down-regulated and 24 metabolites were up-regulated in SM after digestion, which may be because the SM contained the least O-alkyls and anomerics of carbohydrates compared to CHM and CM, and thus its organic matter was more resistant to microbial attack [28]. Significant variations in short-chain amine and the other short-chain organic products were observed after the AD process of CHM, which may be due to higher contents of organic matter (COD) and nitrogen (Table 1) compared with the other manures. Furthermore, the CHM contained more organic nitrogen (as undigested

proteins and uric acid) that hydrolyzed and released large amounts of carbohydrates [29]. The indole of CHM can be converted to produce indole derivatives via the production of intracellular or extracellular enzymes by anaerobic bacterial strains. While long-chain cyclic amines and unsaturated olefin organic products were significantly formed during the AD process of CM, which is mainly due to lignin hydrolysis and cellulose conversion [30,31]. Moreover, the biodegradation of lignin is rate-limiting as the complex of lignin, cellulose, and hemicellulose prevents the conversion of these compounds to the precursor of biogas [32]. This ultimately leads to a lower rate of AD and changes the metabolite profile in CM.

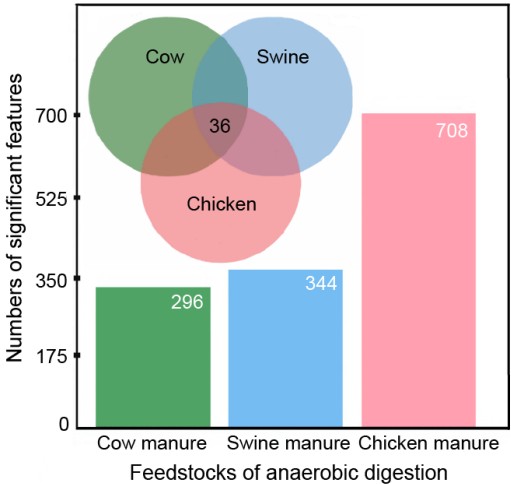

**Figure 2.** Venn diagram of the significant features from the comparison between manure types and their digestates (min FC ≥ 2, max *p*-value ≤ 0.05, max Intensity ≥ 0).

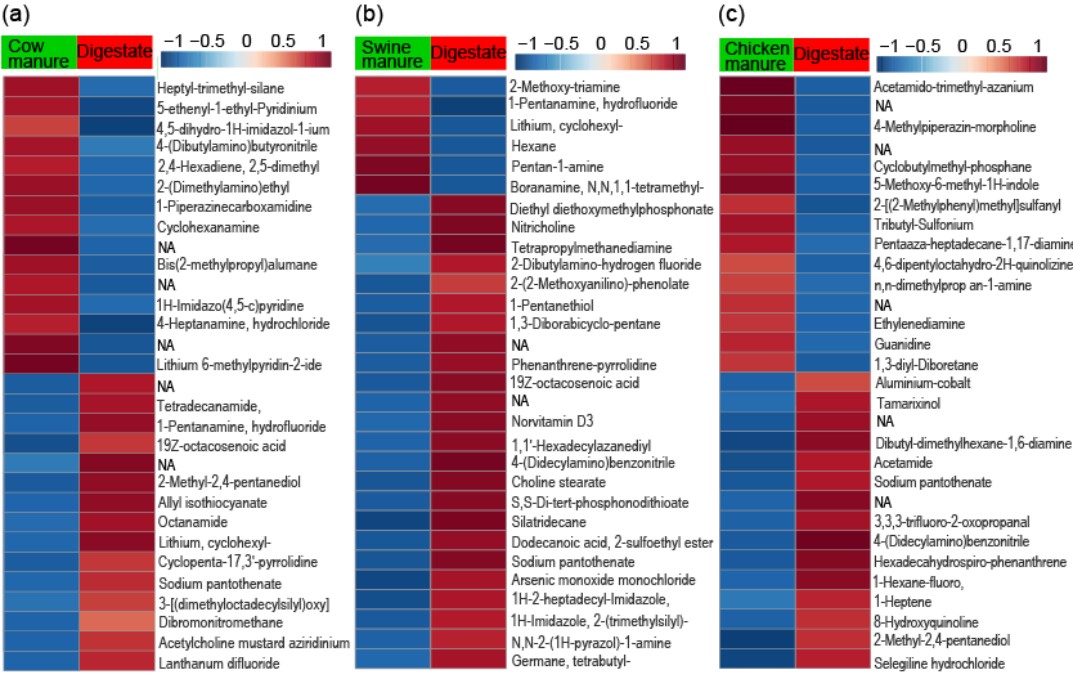

**Figure 3.** Interactive, sortable heat map with customized metabolomic data; (**a**) cow manure and digestate, (**b**) swine manure and digestate, (**c**) chicken manure and digestate ($p < 0.05$).

Furthermore, the manure properties, animal health, digestibility, animal age, production system, protein and fiber contents of the diet, feed digestibility, and environmental condition also influence the metabolite profile during AD of manure [33]. Yang, Fan [34] performed a metabolomic analysis of the microbial community during two-stage AD of corn stalks using GC-MS and identified 23 metabolites

generated during the process. Additionally, they reported that the level of sugars and sugar alcohols increased during methanogenesis and that the fatty acid content increased during acidogenesis. Another study conducted an untargeted metabolomics analysis of *Mucor racemosus* (Douchi) fermentation using GC-MS and identified 114 metabolites [35]. In our study, we used a more systematic, rapid, and efficient technique (XCMS online and MetaboAnalyst programs) for untargeted global profiling of metabolites and their metabolic pathways.

*3.2. Primary Metabolite Derivative Pathways*

The MetaboAnalyst 4.0 database was used to identify the different derivative pathways responsible for primary metabolite generation during AD of CHM, SM, and CM (Table S2). Eleven metabolic pathways were identified that involved the generation of primary metabolites during AD of CHM. The metabolic pathways were tryptophan metabolism, aspartate and asparagine metabolism, tyrosine metabolism, linoleate metabolism, hexose phosphorylation, vitamin B5-coenzyme A (COA) biosynthesis from pantothenate, androgen and estrogen biosynthesis, vitamin B9 (folate) metabolism, COA metabolism, pyrimidine metabolism, C21-steroid hormone biosynthesis, and metabolism. Similarly, we identified hexose phosphorylation, vitamin E metabolism, glycerophospholipid metabolism, and C21-steroid hormone biosynthesis and metabolism pathways, which were responsible for primary metabolite production during AD of CM. The primary metabolite production during AD of SM involved seven pathways: lysine degradation, arginine, and proline metabolism, tyrosine metabolism, tryptophan metabolism, beta-alanine metabolism, D-arginine, and D-ornithine metabolism, and pantothenate and COA biosynthesis pathways.

Generally, the pathways involved amino acid metabolism, hormone metabolism, and vitamin metabolism, producing bioactive compounds in various forms such as peptides, proteins, alkaloids, vitamins, and coenzymes, as well as hormones (Table S2). However, the difference of the chemical derivative pathway during the AD process of CHM CM and SM may be due to the different feed demand, and feed utilization between monogastric animals (SM CHM) and ruminant animals (CM) [36]. In chickens, linoleate is synthesized from the ingested feed and the synthesized linoleate is used for the synthesis of polyunsaturated fatty acids [37]. Similarly, hexose sugars are synthesized, which must be phosphorylated before their catabolism by chickens [38]. The vitamin, pantothenic acid is the functional group of coenzyme A, which is generally added to the diet of monogastric animals, like chicken and swine [39]. Additionally, pantothenic acid can also be synthesized by the rumen microorganisms in the ruminants [40]. Similarly, vitamin B is added to the diet of monogastric animals (chicken and swine). However, rumen microbes can synthesize vitamin B in ruminants (cattle) [41]. Tryptophan is the preferred material for decomposition in CHM and CM. Furthermore, the difference in chemical change during the AD process of CHM, CM, and SM may be due to the higher nutrient availability (Table 1) and rate of decomposition in the manure of monogastric animals when compared to the manure of ruminants (CM), which have a higher content of lignin, a phenolic compound [36,42,43]. In ruminants, the partially digested food in the stomach is returned to the mouth, regurgitated with saliva, and then enters the stomach for second digestion. Hence, CM contains less easy-to-utilize amino acids such as tryptophan and a higher content of lignin basic structure substances [44,45].

Furthermore, the gastrointestinal tract conditions (e.g., redox potential of −350 to −150 mV, temperature of 36–39 °C, and pH of 5.5–7.0) of ruminants are favorable for the microbial hydrolysis and digestion of lignocellulosic biomass [46]. Li et al. [47] reported that metabolites such as indole acetic acid (IAA), skatole, and indole, are produced from L-tryptophan metabolism during AD of dairy manure. There are two metabolic pathways involved in L-tryptophan metabolism. L-tryptophan can be converted to skatole via IAA or it is directly converted to indole. Xu et al. [48] used LC-MS for non-target analysis of metabolite profiles during microbial fermentation of Fu brick tea. They reported that catechin degradation pathways, including B ring fission and L-theanine derivative pathways, are involved in the production of metabolites. Previously, either GC-MS or LC-MS alone was employed for the identification of few metabolites and their derivative pathways. However, we have used XCMS

and metabolomic programs to comprehensively evaluate the metabolite profile of the digestate from manure (swine, chicken, and cattle) and their multiple derivative pathways. Therefore, the higher nutrient contents and contents of metabolites in the manure from monogastric animals like chicken and swine compared with ruminants (cattle) suggested their greater potential as organic fertilizer to improve plant growth, development, and production as well as their ability to withstand different diseases.

### 3.3. Characterization of Bioactive Compounds

The bioactive compounds detected in the manure and their digestate are presented in Figure 4. Four amino acids, namely, L-arginine, L-ornithine, L-cysteine, and L-aspartate were detected in both CHM and its digestate, which are mainly derived from aspartate, asparagine, tyrosine, pyrimidine, and pantothenate metabolic pathways, respectively (Table S2). All amino acids were up-regulated after AD of CHM (Figure 4c). Four amino acids, D-arginine, carnitine, L-arginine, L-ornithine were detected in the SM and its digestate, which are produced from lysine, D-arginine, and D-ornithine metabolic pathways, respectively. The content of all four amino acids was up-regulated after AD (Figure 4b). The hormones, L-adrenaline, 19-hydroxy androstenedione, and estrone were mainly detected in CHM (Figure 4c). The androgens can be transformed into steroids during digestion. The AD of manure can decompose steroid hormones and decrease the environmental risks of hormones to aquifers and living organisms associated with land application of manure [49].

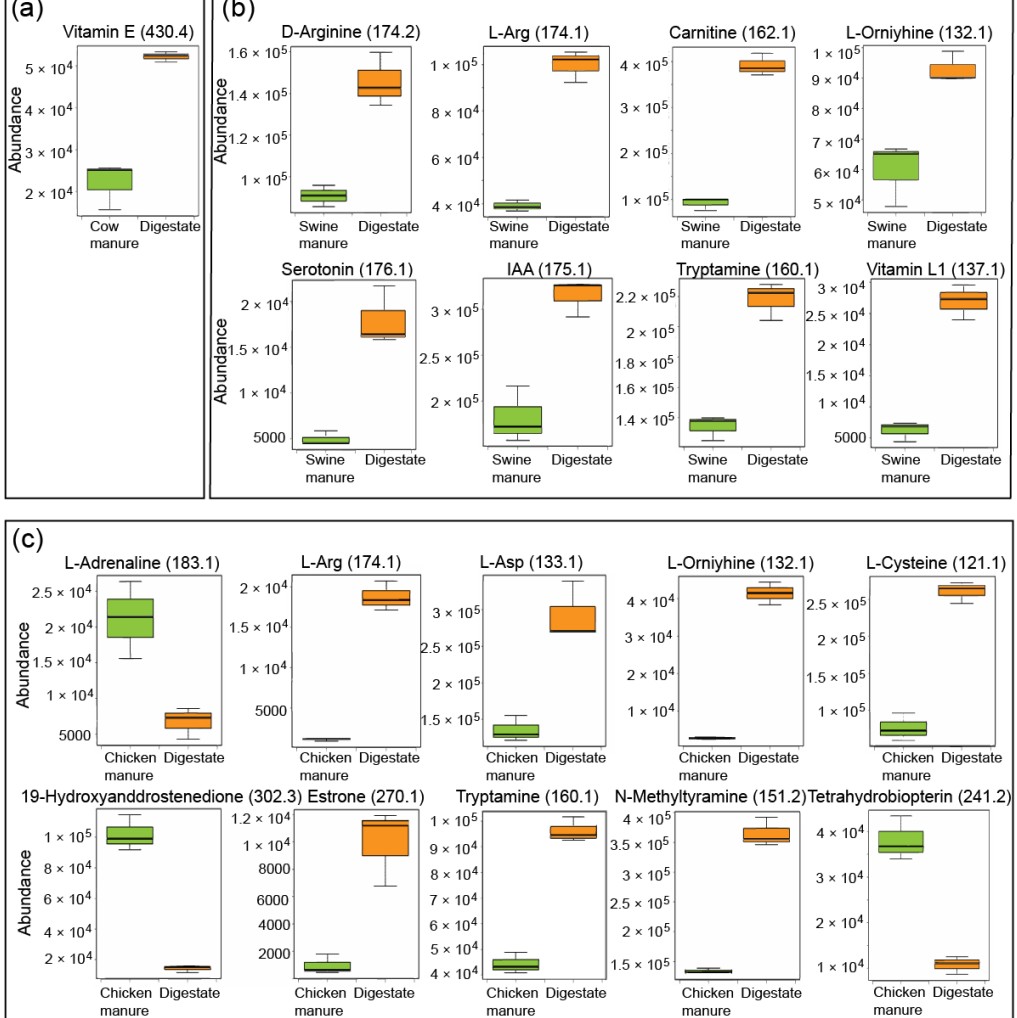

**Figure 4.** Comparison of the relative content of bioactive compounds; (**a**) cow manure and digestate, (**b**) swine manure and digestate, (**c**) chicken manure and digestate.

The content of alkaloids, such as tryptamine, serotonin, indole acetic acid (IAA) increased after AD of SM (Figure 4b). Similarly, the content of alkaloids, such as tryptamine, N-methyltyramine, and serotonin, was up-regulated after AD of CHM (Figure 4c). The alkaloids in CHM and SM are generated through the tryptophan and tyrosine metabolic pathways (Table S2). Tryptophan is markedly decomposed during AD, and it is metabolized to generate metabolites like indole, indolic acid, alkaloids, skatole and tryptamine [50]. Moreover, indoles, alkaloids, and vitamins produced through anoxic metabolism of tryptophan can also be produced by the microbes in the rumen, ceca, and colon of monogastric animals. The related microorganism continues to expand after entering the anaerobic reactors with comparable conditions [51]. Hence, the alkaloids and vitamins produced from easy-to-utilize amino acids like tryptophan and tryptamine in CHM and SM were enriched through AD. However, CM has a low content of easy-to-utilize amino acids and a high content of lignin basic structure substances [44,45]. Vitamins, such as tetrahydrobiopterin vitamin B5, vitamin B9, vitamin E, and vitamin L, were detected in CHM, CM, SW, and their digestate (Figure 4). They are mainly produced through the tryptophan, tyrosine, and pantothenate biosynthetic pathways.

These bioactive compounds are reported to have antifungal and antimicrobial activities. Hence, identifying and characterizing these bioactive compounds is vital to evaluate the effect of digestate land application on controlling phytopathogens. Lu et al. [21] identified some bioactive compounds using GC-MS in anaerobically digested chicken slurry and evaluated their antifungal activity against *Fusarium oxysporum*. They concluded that bioactive compounds in chicken slurry digestate markedly reduced the growth of *Fusarium oxysporum*. Similarly, alkaloids and their derivatives were also reported to exhibit antimicrobial activity [12]. Likewise, the existence of bioactive compounds with variable content of anaerobic digestate of different manures was confirmed in the current study along with the metabolic pathways of their generation. Most of these bioactive compounds have antimicrobial activity against various pathogens. Therefore, more studies are needed for exploiting the antimicrobial potential of digestate against various plant diseases in addition to their ability to improve plant growth and soil fertility.

## 4. Conclusions

In this study, we performed untargeted metabolite profiling of chicken, cattle and swine manure before and after AD by integrating the online XCMS and MetaboAnalyst programs. We identified 479, 241, and 431 metabolites after the AD of CHM, CM, and SM, respectively. Comparing the difference in chemical compositions of digestate from the monogastric and ruminant manure after AD. Additionally, 11, 4, and 7 metabolic pathways were identified to be involved in the production of metabolites during the AD process of chicken, cattle, and swine manure, respectively. The bioactive compounds of digestate from different manure were identified. The findings of the current study may help in elucidating the underlying mechanism of antimicrobial activity, protection against disease, and change in plant metabolite profile (primary and secondary) induced by land application of manure digestate.

**Supplementary Materials:** The following are available online at http://www.mdpi.com/2073-4441/11/11/2420/s1, Table S1: The shared significant features from paired-comparison between manures and digestates, Table S2: Metabolic pathways involved in the formation of bioactive compounds during anaerobic digestion of chicken, cow and pig manure.

**Author Contributions:** Conceptualization J.L.; methodology J.L.; software, J.L.; formal analysis A.M.; investigation, J.M.; resources, J.D.; data curation, W.C.; writing—original draft preparation J.L.; writing—review and editing A.M.; visualization J.L.; supervision A.M.; project administration, R.D.; funding acquisition R.D.

**Funding:** This research received no external funding.

**Conflicts of Interest:** The authors declare no conflicts of interest.

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
