# Peer review of "Untargeted Metabolite Profiling for Screening Bioactive Compounds in Digestate of Manure under Anaerobic Digestion"

_water, doi:10.3390/w11112420_

Round 1

Reviewer 1 Report

This paper deals with a very important aspect of digestate influence on the crops. The authors made an attempt to find the bioactive compounds in chicken, swine and cattle manure and their digestate produced through anaerobic digestion. The experiments were properly planned and realized. The data analysis included novel methods. I recommend minor revision. 

Specific comments:

Page 10 Lines 301 and 302: Latin names of microbes should be written in italic.

Author Response

Response: Bundle of thanks for your comments

Point 1: [1] Page 10 Lines 301 and 302: Latin names of microbes should be written in italic.

Response: The names of the microbes are italicized as per suggestion [Page#10; Line 321-322]

Reviewer 2 Report

Lu et al. Water review

This MS used metabolite profiling to examine the metabolites present in chicken, swine and cattle manure before and after anaerobic digestion at biogas plants in China. The authors are interested in identifying bioactive compounds in the digestates that might be used to improve disease resistance and yield in crop plants fertilized with them.

The authors identified several compounds that appear to be present in significantly different amounts in digestates versus feedstock manure. The paper however lacks clarity in how the results were obtained (e.g., quantification of bioactive compounds) and interpreted statistically, and doesn’t do a good job in linking their results back to their primary objectives of using digestates from biogas plants to improve crop plant health. Table 3 is also missing from the MS (referred to in line 223), which makes it hard to follow their discussion about pathways that are predicted to be upregulated based on their metabolite levels. Considering the scope of the journal on water quality, there is no discussion about the impact of the AD process on water quality in agricultural areas where the AD digestate is sprayed or in receiving waters where it’s released into the environment.

For these reasons, I recommend a major revision prior to publication in Water. Specific comments are below.

Line 40: ...”promotes disease resistance in plants” needs a reference.

M&M 2.1: need more info on how you chose the biogas plants. Convenience, differences in processes that allowed sampling to find statistically significant differences in common/distinct metabolites, etc.

L88: “...given in table 1”: need detail on how you measured the endpoints in the table like VS, COD, etc. Define all of the acronyms in the table in the table legend. N=? for each measurement? Are values means +/- SD or SEM?

L91: “the anaerobic digestate”: how much was used (e.g., g wet weight).

L93:” and then extracted” for how long?

L98: “repeated three times”... on different days? During what months and over what period of time? Are these n=3 biological or experimental repetitions?

Table 1: You never mention these values again in the text. How do the values in the table influence your discussion about metabolic pathways affected by the AD process? E.g., it’s interesting that AD is not effective at reducing the toxicity of the manure, which would be a problem if the AD digestate was released into waterways.

L104: “1.5 uL of extract”: how did you standardize each sample injected into the MS to allow quantification of fold changes in metabolite levels in the solid wastes (e.g., as pmol/g wet or dry weight of manure/digestate.

L126: Need more info on how significant changes in metabolite levels were identified. Does “threshold >= 1.5” mean that metabolites in digestate whose amounts changed by +/-1.5-fold versus the feedstock manure were termed significantly different? Is this -fold change commonly used to determine significance in metabolomics?

L149: upregulated; L151 downregulated: are all these >=1.5-fold changed versus levels in manure?

L156: not “CM owned” but “CM had”

L164: “manure”: from what animal?

Fig 1: how did you account for your experimental replicates (n=3?) in these plots? How similar were results across experimental reps?

L167: “significant features”; L179: “characteristic features”: do you mean metabolites? Ions? Define.

L173-4: “represents the fold change”: need a colour scale in the figure to tell us what the colour intensities mean.

L184: not “food” but “feed”

L188-9: “the content of _the 36_ common metabolites”

L194: not “which was because” but “which may be because”

L198: “nitrogen organic matter”: give an example of such a compound.

L212: “identified 23 metabolites” during what process?

Italicize genus/species names: Mucor racemosus (L215); Fusarium oxysporum (L301-2).

L218-220: “We identified about...”: delete as you said this already in  L153.

L222: different “metabolic”, not “derivative” pathways

L223: Table 3 missing from MS

Section 3.2: need to link discussion of metabolites and pathways and the implications these have on the suitability of AD digestates as plant fertilizers better in this section. Same with linking discussion of animal digestion back to the metabolite profiles you found.

L242: “...before _their catabolism_ by chicken”

L247-8: “tryptophan is the preferred material for decomposition in CHM and CM”: what do you mean?

L268: adrenaline/epinephrine is not an AA but an AA-derived hormone.

L267: “compounds detected”: were these at significantly different values, and that’s why there are only a small number of compounds quantified in Fig 4?

Fig 4: how is “abundance” of these compounds quantified? Are these relative values? If not, these should be expressed as e.g., pmol / g wet weight of manure/digestate. Are _all_ of the compounds that were sig diff between manures and digestates present here? Did you run statistics to show this (e.g., t-tests?) What was the cutoff p-value for significance? Is each sample n-3? Are these standard boxplots?

L297: “_Some_ bioactive compounds...” is more accurate.

Section 3.3: You talk a lot about Trp values changing after AD and how this will impact the abundance of compounds derived from and related to Trp, but Trp is not among the compounds you found to be significantly different between any of the manures and digestates in Fig. 4. This is perhaps surprising, and you should discuss.

L303: the discussion peters out without coming to any firm conclusions; e.g., whether the AD digestates might be effective in increasing plant disease resistance and yield. For example, you should discuss concentrations of the “bioactive compounds” you identified as being present after AD and whether they might be high enough to protect plants onto which the digestates are sprayed. What are next steps to take to show this (e.g., working with plant physiologists to test the effect of relevant AD concentrations when applied to crops on yield and plant health).

L311: bioactive compounds were not “screened”, but “identified”.

Author Response

Response: Bundle of thanks for your valuable comments

Point 1: [1] Line 40: ...”promotes disease resistance in plants” needs a reference.

Response 1: The reference is provided [Page#1, Line 41]

Point 2: [2] M&M 2.1: need more info on how you chose the biogas plants. Convenience, differences in processes that allowed sampling to find statistically significant differences in common/distinct metabolites, etc.

Response 2: Thanks for suggestion. The information regarding the selection of the biogas plants for sampling has been provided [Page#2, Line 84-87].

Point 3: [3] L88: “...given in table 1”: need detail on how you measured the endpoints in the table like VS, COD, etc. Define all of the acronyms in the table in the table legend. N=? for each measurement? Are values means +/- SD or SEM?

Response 3: A brief details of the analytical methods used for the parameters given in Table 1 has been incorporated. The acronyms are also described in the table legend. Yes the values are presented as Mean±SD [Page# 2, 3; Line 90-95; 110-111]

Point 4: [4] L91: “the anaerobic digestate”: how much was used (e.g., g wet weight).

Response 4: 15 ml of digestate was mixed with 15 ml extractant (1:1) for extraction using ultrasound. The same is provided in the main text of the article [Page#3, Line 100-101]

Point 5: [5] L93:” and then extracted” for how long?

Response 5: Extraction was done for 20 minutes. The same has been incorporated in the text [Page#3; Line 103]

Point 6: [6] L98: “repeated three times”... on different days? During what months and over what period of time? Are these n=3 biological or experimental repetitions?

Response 6: These are the experimental repeats and were done on the same day.

Point 7: [7] You never mention these values again in the text. How do the values in the table influence your discussion about metabolic pathways affected by the AD process? E.g., it’s interesting that AD is not effective at reducing the toxicity of the manure, which would be a problem if the AD digestate was released into waterways.

Response 7: Thanks for pointing out. The data given in Table 1 is described in the text while discussing metabolites [Page#7, 8; Line 211-212, 264]

Point 8: [8] L104: “1.5 uL of extract”: how did you standardize each sample injected into the MS to allow quantification of fold changes in metabolite levels in the solid wastes (e.g., as pmol/g wet or dry weight of manure/digestate?

Response 8: The extracts from the manures and digestates samples (1.5 µL) was injected manually using a standardized microsyringe provided by the company after appropriate quality assurance and claimed that the microsyringe will inject standard amount e.g., 1.5 µL.

Point 9: [9] L126: Need more info on how significant changes in metabolite levels were identified. Does “threshold >= 1.5” mean that metabolites in digestate whose amounts changed by +/-1.5-fold versus the feedstock manure were termed significantly different? Is this -fold change commonly used to determine significance in metabolomics?

Response 9: XCMS is web based platform employed for untargeted profiling of metabolites and identifies features whose relative intensity varies between sample groups and calculates p-values as well as fold changes. Fold change represents the difference in relative intensity between group and it is an important parameter to characterize the degree of metabolite change. Moreover, statistical significance of fold change is calculated by Welch t- test. The lower fold change thresholds (1.5-2.0) have been suggested to be used for the samples of animal origin in the literature (Crews et al., 2009; Massaon et al., 2011). So based on the literature the threshold fold change was chosen in the current study.

References

Crews B, et al. Variability analysis of human plasma and cerebral spinal fluid reveals statistical significance of changes in mass spectrometry-based metabolomics data. Analytical chemistry. 2009; 81:8538–8544.

Masson P, Spagou K, Nicholson JK, Want EJ. Technical and biological variation in UPLC-MS based untargeted metabolic profiling of liver extracts: application in an experimental toxicity study on galactosamine. Anal Chem. 2011; 83:1116–1123.

Point 10: [10] L149: upregulated; L151 downregulated: are all these >=1.5-fold changed versus levels in manure?

Response 10: Yes the up-regulated and down-regulated metabolites are those metabolites whose intensity increase or decrease during AD process. How much their intensity is increase/decrease is based on the fold change threshold and represented by the size of circle in the Figure 1.

Point 11: [11] L156: not “CM owned” but “CM had”

Response 11: The suggested improvement is made [Page#4, Line 168]

Point 12: [12] L164: “manure”: from what animal?

Response 12: The sentence has been modified to remove ambiguity [Page#4, Line 175]

Point 13: [13] Fig 1: how did you account for your experimental replicates (n=3?) in these plots? How similar were results across experimental reps?

Response 13: The detail about the analysis using XCMS online is already provided in the response of the comment 9.

Point 14: [14] L167: “significant features”; L179: “characteristic features”: do you mean metabolites? Ions? Define

Response 14: The term “Characteristic features” is defined for clarification as suggested [Page#5, Line 178]

Point 15: [15] L173-4: “represents the fold change”: need a colour scale in the figure to tell us what the colour intensities mean.

Response 15: Thanks for suggestion but is difficult to provided colour scale because is it software generated Figure

Point 16: [16] L184: not “food” but “feed”

Response 16: The term “food” is replace by “feed” [Page#5 Line 194]

Point 17: [17] L188-9: “the content of _the 36_ common metabolites”

Response 17: Suggested improvement is made [Page#6, Line 201]

Point 18: [18] L198: “nitrogen organic matter”: give an example of such a compound.

Response 18: The sentence is modified by incorporating the suggested example and to remove ambiguity [Page#6, Line 212-213]

Point 19: [19] L212: “identified 23 metabolites” during what process?

Response 19: Suggested change is made by providing the name of the process [Page#7, Line 228]

Point 20: [20] Italicize genus/species names: Mucor racemosus (L215); Fusarium oxysporum (L301-2).

Response 20: The names of the microbes are italicizes as suggested [Page#7, 10, Line 231, 321-322]

Point 21: [21] L218-220: “We identified about...”: delete as you said this already in  L153.

Response 21: The suggested improvement is made by deleting the sentences [Page#7; Line 234-235]

Point 22: [22] L223: Table 3 missing from MS

Response 22: Sorry for mistake. The Table 3 is given as Table S2 in the supplementary materials.

Point 23: [23] Section 3.2: need to link discussion of metabolites and pathways and the implications these have on the suitability of AD digestates as plant fertilizers better in this section. Same with linking discussion of animal digestion back to the metabolite profiles you found.

Response 23: The discussion in section 3.2 is linked to the land application of digestate and suitability for improving crop production as well as enhancing plant health [Page#8; Line 282-285]

Point 24: [24] L242: “...before _their catabolism_ by chicken”

Response 24: The suggested improvement is made [Page#8; Line 257]

Point 25: [25] L247-8: “tryptophan is the preferred material for decomposition in CHM and CM”: what do you mean?

Response 25: Tryptophan being an important dietary supplement that is used as substrate for the protein synthesis in the animal body. Moreover, it is metabolized to produce various bioactive compounds like indoles, serotonin and others. For this reason it may be responsible for the detection of metabolites and bioactive compounds in manure and digestates. The detail about tryptophan is given in the comment 30.

Point 26: [26] L268: adrenaline/epinephrine is not an AA but an AA-derived hormone.

Response 26: Thanks for point out. The mistake is corrected [Page 1, 8; Line 20-21; 288, 291, 295]

Point 27: [27] L267: “compounds detected”: were these at significantly different values, and that’s why there are only a small number of compounds quantified in Fig 4?

Response 27: Yes these are the most significant and biologically active compounds among large number of detected compounds.

Point 28: [28] Fig 4: how is “abundance” of these compounds quantified? Are these relative values? If not, these should be expressed as e.g., pmol / g wet weight of manure/digestate. Are _all_ of the compounds that were sig diff between manures and digestates present here? Did you run statistics to show this (e.g., t-tests?) What was the cutoff p-value for significance? Is each sample n-3? Are these standard boxplots?

Response 28: Yes the “abundance” here is the value for relative abundance. Yes, the manures  and their corresponding anaerobic digestate were analyzed as pairwise job to perform an unpaired parametric t-test (Welch test)  and the statistically significant features were identified using p-value ≤ 0.05 and threshold ≥ 1.5 as shown in Fig1. Each sample is n=3. These are the Box-whisker plots generated by XCMS online for a multigroup analysis.

Point 29: [29] L297: “_Some_ bioactive compounds...” is more accurate.

Response 29: The suggested change is made by changing “various” to “some” [Page#9; Line 320]

Point 30: [30] Section 3.3: You talk a lot about Trp values changing after AD and how this will impact the abundance of compounds derived from and related to Trp, but Trp is not among the compounds you found to be significantly different between any of the manures and digestates in Fig. 4. This is perhaps surprising, and you should discuss.

Response 30: Tryptophan is one of the important dietary supplements that can alter intestinal microbial composition and diversity. Besides serving as a substrate for protein synthesis, Tryptophan (Trp) is metabolized to a variety of biologically active compounds, such as serotonin, tryptamine, indoles etc., (Yao et al., 2011; Cervenka et al., 2017). Moreover, intestinal commensal bacteria can actively metabolize Trp to produce IAA and indole, two critical metabolites that enhance both intestinal mucosal barrier integrity and immune function of the organisms via activation of the related signalling pathways (Zelante et al., 2013; Jin et al., 2014; Venkatesh et al., 2014; Lamas et al., 2016; Barratt et al., 2017). Therefore, Trp a functional amino acid that regulates animal’s physiology and metabolism and supplemented in the feed. Moreover, the biological active compounds like serotonin, tryptamine, indoles and other generated through metabolism of Trp may release out of the animal’s body as manures. Being important feed supplement and its role in the various physiological functions it is discussed as reason for detection of the compounds that originate from its metabolism.

References

Barratt, M. J., Lebrilla, C., Shapiro, H. Y., and Gordon, J. I. (2017). The gut microbiota, food science, and human nutrition: a timely marriage. Cell Host Microbe 22, 134–141.

Cervenka, I., Agudelo, L. Z., and Ruas, J. L. (2017). Kynurenines: tryptophan’s metabolites in exercise, inflammation, and mental health. Science 357

Jin, U. H., Lee, S. O., Sridharan, G., Lee, K., Davidson, L. A., Jayaraman, A., et al. (2014). Microbiome-derived tryptophan metabolites and their aryl hydrocarbon receptor-dependent agonist and antagonist activities. Mol. Pharmacol. 85, 777–788.

Lamas, B., Richard, M. L., Leducq, V., Pham, H. P., Michel, M. L., Da Costa, G.,et al. (2016). CARD9 impacts colitis by altering gut microbiota metabolism of tryptophan into aryl hydrocarbon receptor ligands. Nat. Med. 22, 598–605.

Yao, K., Fang, J., Yin, Y. L., Feng, Z. M., Tang, Z. R., and Wu, G. (2011). Tryptophan metabolism in animals: important roles in nutrition and health. Front. Biosci. 3, 286–297.

Zelante, T., Iannitti, R. G., Cunha, C., De Luca, A., Giovannini, G., Pieraccini, G., et al. (2013). Tryptophan catabolites from microbiota engage aryl hydrocarbon receptor and balance mucosal reactivity via interleukin-22. Immunity 39, 372–385.

Point 31: [31] L303: the discussion peters out without coming to any firm conclusions; e.g., whether the AD digestates might be effective in increasing plant disease resistance and yield. For example, you should discuss concentrations of the “bioactive compounds” you identified as being present after AD and whether they might be high enough to protect plants onto which the digestates are sprayed. What are next steps to take to show this (e.g., working with plant physiologists to test the effect of relevant AD concentrations when applied to crops on yield and plant health).

Response 31: The discussion is improved by incorporating conclusion and suggestion for future studies. [Page#10; Line 324-329]

Point 32: [32] L311: bioactive compounds were not “screened”, but “identified”.

Response 32: Thanks for suggestion. The word “screened” is replace with “identified” [Page#10; Line 337]

Reviewer 3 Report

Dear Authors,

Thank you very much for the possibility of reading this article.

I have carefully read your manuscript. In my opinion, this manuscript is very interesting and valuable. This paper studies untargeted metabolite profiling of chicken, cattle and swine manure before and after anaerobic digestion (AD) by integrating online XCMS and MetaboAnalyst programs. The Authors identified 479, 241, and 431 metabolites after the AD and 11, 4, and 7 metabolic pathways which are involved in the production of metabolites during AD process of chicken, cattle and swine manure, respectively. The bioactive compounds of digestate identified by the Authors will help explain mechanism of promoting disease resistance by the digestate. In my opinion, the results obtained by the Authors are unique and very useful for further deeper biochemical and physiological analyzes.

Detailed remarks:

Line 170 – In my opinion the sentence: “Green colored spots ...” can be deleted because this information is above.

Lines 223 and 237 – there is a reference to the results in Table 3 – there is no Table 3 in the manuscript.

Best regards,

Reviewer

Author Response

Response: Bundle of thanks for your valuable comments

Point 1: [1] Line 170 – In my opinion the sentence: “Green colored spots ...” can be deleted because this information is above.

Response1: The suggested improvement is made by deleting the sentence [Page#5; Line 180-181]

Point 2: [2] Lines 223 and 237 – there is a reference to the results in Table 3 – there is no Table 3 in the manuscript.

Response2: We are sorry for the mistake. The Table 3 is given as Table S2 in the supplementary materials.

Round 2

Reviewer 2 Report

Most of the changes I suggested were implemented, so the MS is now suitable for publication.

I suggest a few changes first, as below:

Line 18-9: Twelve, 8 and 5 metabolic pathways _were affected by the AD process in_ CHM, SM,...

Table 1, define CSTR and USR in legend. Indicate n=? for the measured endpoints.

The MS needs to be lightly edited for English prior to publication.

Author Response

We are obliged to the reviewers of this manuscript, which contributed to improve its quality. The response file explaining how the authors have dealt with the specific referees’ comments are integrated in this reviewer’s comments (in bold red). Relevant changes (highlighted) were introduced throughout the manuscript text in track change form.

Point 1: [1] Line 18-9: Twelve, 8 and 5 metabolic pathways _were affected by the AD process in_ CHM, SM,...

Response: Bundle of thanks for suggestion. The suggested improvement is made.

Point 2: [2] Table 1, define CSTR and USR in legend. Indicate n=? for the measured endpoints.

Response: The terms CSTR and USR are defined in the Table 1 legend as per suggestion. Moreover, the number of the samples analyzed is incorporated in the title of Table 1

Point 3: [3] The MS needs to be lightly edited for English prior to publication.

Response: The English is corrected and highlighted in track form throughout the article
